# Spectral Self-supervised Feature Selection

**Daniel Segal**                      *segaldaniel23@gmail.com*
*Hebrew University*

**Ofir Lindenbaum**                      *ofirlin@gmail.com*
*Bar Ilan University*

**Ariel Jaffe**                    *ariel.jaffe@mail.huji.ac.il*
*Hebrew University*

**Reviewed on OpenReview:** *https://openreview.net/forum?id=t0EJiOd9Lg*

## Abstract

Choosing a meaningful subset of features from high-dimensional observations in unsupervised settings can greatly enhance the accuracy of downstream analysis tasks, such as clustering or dimensionality reduction, and provide valuable insights into the sources of heterogeneity in a given dataset. In this paper, we propose a self-supervised graph-based approach for unsupervised feature selection. Our method's core involves computing robust pseudo-labels by applying simple processing steps to the graph Laplacian's eigenvectors. The subset of eigenvectors used for computing pseudo-labels is chosen based on a model stability criterion. We then measure the importance of each feature by training a surrogate model to predict the pseudo-labels from the observations. Our approach is shown to be robust to challenging scenarios, such as the presence of outliers and complex substructures. We demonstrate the effectiveness of our method through experiments on real-world datasets from multiple domains, with a particular emphasis on biological datasets.

## 1 Introduction

Improvements in sampling technology enable scientists across many disciplines to acquire numerous variables from biological or physical systems. One of the critical challenges in real-world scientific data is the presence of noisy, information-poor, or nuisance features. While such features could be mildly harmful to supervised learning, they could dramatically affect the outcome of downstream analysis tasks (e.g., clustering or manifold learning) in unsupervised settings (Mahdavi et al., 2019). There is thus a growing need for unsupervised feature selection schemes that enhance latent signals of interest by removing nuisance variables and thus advance reliable data-driven scientific discovery.

Unsupervised Feature Selection (UFS) methods are designed to identify a set of informative features that can improve the outcome of downstream analysis tasks such as clustering and manifold learning. With the lack of labels, however, selecting features becomes highly challenging since the downstream task cannot be used to drive the selection of features. As an alternative, most UFS methods use a label-free criterion that correlates with the downstream task. For instance, many UFS schemes rely on a reconstruction prior (Li et al., 2017) and seek a subset of features that can be used to reconstruct the entire dataset as accurately as possible. Several works use Autoencoders (AE) to learn a reduced representation of the data while applying a sparsification penalty to force the AE to remove redundant features. This idea was implemented with several types of sparsity-inducing regularizers, including $\ell_{2,1}$ based (Chandra and Sharma, 2015; Han et al., 2018), relaxed $\ell_0$ (Balın et al., 2019; Shaham et al., 2022; Svirsky and Lindenbaum) and more.

One of the most commonly used criteria for UFS is feature smoothness. A common hypothesis is that the structure of interest, such as clusters or a manifold, can be captured using a graph representation(Ng et al., 2001). The smoothness of features is measured using the Laplacian Score (LS) (He et al., 2005), which is

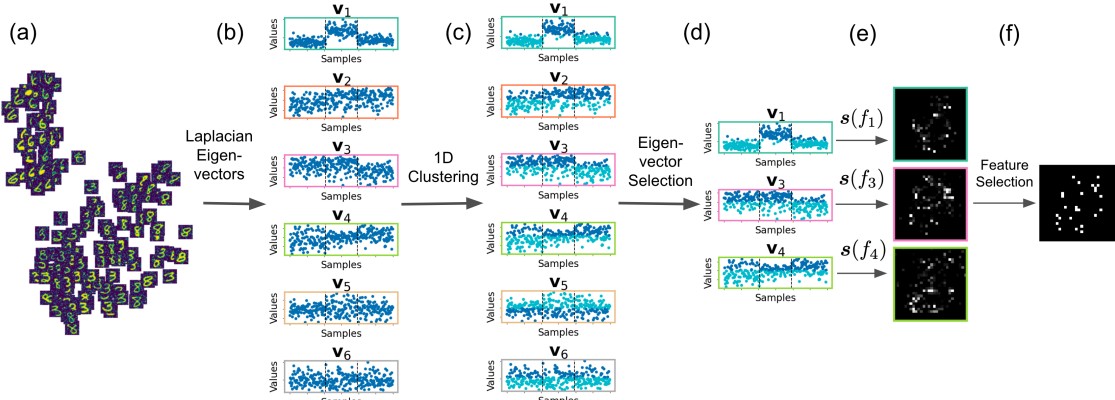

Figure 1: Illustration of SSFS. Panel (a) shows a tSNE scatter plot of noisy MNIST digits (3, 6, 8). Panel (b) presents the six leading eigenvectors computed based on the graph Laplacian of the data. Samples are ordered according to the identity of the digit. (c) We apply the $k$-medoids algorithm to compute pseudo-labels $\boldsymbol{y}_i^*$. These are presented as colors overlayed on the eigenvectors. (d) We select the three eigenvectors whose pseudo-labels are the most "stable" with respect to several prediction models (see Section 3.2). (e) For each data feature we estimate its importance score for each of the selected eigenvectors (see Section 3.3). (f) We aggregate the feature scores across eigenvectors.

equal to the Rayleigh quotient of a normalized feature vector and the graph Laplacian. A feature that is smooth with respect to the graph is considered to be associated with the primary underlying data structures. There are many other UFS methods that use a graph to select informative features Li et al. (2018); Roffo et al. (2017); Zhu et al. (2017; 2020); Xie et al. (2023). (Li et al., 2012) derived Nonnegative Discriminative Feature Selection (NDFS), which performs feature selection and spectral clustering simultaneously. Its extension Li and Tang (2015) adds a loss term to prevent the joint selection of correlated features.

Embedded unsupervised feature selection schemes aim to cluster the data while simultaneously removing irrelevant features. Examples include Wang et al. (2015), which performs the selection directly on the clustering matrix, and Zhu and Yang (2018), which learns feature weights while maximizing the distance between clusters. In recent years, several works have derived self-supervised learning methods for feature selection. The key idea is to design a supervised type learning task with pseudo-labels that do not require human annotation. A seminal work based on this paradigm is Multi-Cluster Feature Selection (MCFS) (Cai et al., 2010). MCFS uses the eigenvectors of the graph Laplacian as pseudo-labels and learns the informative features by optimizing over an $\ell_1$ regularized least squares problem. More recently, Lee et al. (2021) used self-supervision with correlated random gates to enhance the performance of feature selection.

In this work, we present a spectral self-supervised scheme for feature selection. Our main innovation is to properly select a subset of Laplacian eigenvectors by a multi-stage approach. Firstly, we generate robust discrete pseudo-labels from the eigenvectors and filter them based on a stability measure. Next, we fit flexible surrogate classification models on the selected eigenvectors and query the models for feature scores. Using these components, we can identify informative features that are effective for clustering on real-world datasets.

## 2 Preliminaries

### 2.1 Laplacian score and representation-based feature selection

Generating a graph-based representation for a group of high-dimensional observations has become a common practice for unsupervised learning tasks. In manifold learning, methods such as ISOMAPS (Tenenbaum et al., 2000), LLE (Roweis and Saul, 2000), Laplacian eigenmaps (Belkin and Niyogi, 2003), and diffusion maps (Coifman and Lafon, 2006) compute a low-dimensional representation that is associated with the manifold's

latent structure. In spectral clustering, a set of points is partitioned by applying the $k$-means algorithm to the leading Laplacian eigenvectors (Ng et al., 2001).

In graph methods, each node $v_i$ corresponds to one of the observations $\boldsymbol{x}_i \in \mathbb{R}^p$. The weight $W_{ij}$ between two nodes $v_i, v_j$ is computed based on some kernel function $K(\boldsymbol{x}_i, \boldsymbol{x}_j)$. For example, the popular Gaussian kernel is equal to,

$$K(\boldsymbol{x}_i, \boldsymbol{x}_j) = \exp\left(-\frac{\|\boldsymbol{x}_i - \boldsymbol{x}_j\|^2}{2\sigma^2}\right).$$

Where the parameter $\sigma$ determines the bandwidth of the kernel function. Let $\boldsymbol{D}$ be a diagonal matrix with the degree of each node in the diagonal, such that $D_{ii} = \sum_j W_{ij}$. The unnormalized graph Laplacian matrix is equal to $\boldsymbol{L} = \boldsymbol{D} - \boldsymbol{W}$. For any vector $\boldsymbol{v} \in \mathbb{R}^n$ we have the following equality (Von Luxburg, 2007),

$$\boldsymbol{v}^T \boldsymbol{L} \boldsymbol{v} = \frac{1}{2} \sum_{i,j} \left(v_i - v_j\right)^2 W_{ij}. \tag{1}$$

The quadratic form in equation 1 gives rise to a notion of graph *smoothness*. (Ricaud et al., 2019; Shuman et al., 2013). A vector is smooth with respect to a graph if it has similar values on pairs of nodes connected with an edge with a significant weight. This notion underlies the Laplacian score suggested as a measure for unsupervised feature selection (He et al., 2005). Let $\boldsymbol{f}_m \in \mathbb{R}^n$ denote the values of the $m$-th feature for all observations. The Laplacian score $s_m$ is equal to,

$$s_m = \boldsymbol{f}_m^T \boldsymbol{L} \boldsymbol{f}_m = \frac{1}{2} \sum_{i,j} \left(f_{m,i} - f_{m,j}\right)^2 W_{ij}. \tag{2}$$

A low score indicates that a feature is smooth with respect to the computed graph and thus strongly associated with the latent structure of the high-dimensional data $\boldsymbol{x}_1, \ldots, \boldsymbol{x}_n$. The notion of the Laplacian score has been the basis of several other feature selection methods as well (Lindenbaum et al., 2021; Shaham et al., 2022; Zhu et al., 2012).

Let $\boldsymbol{v}_i, \lambda_i$ denote the $i$-th smallest eigenvector and eigenvalue of the Laplacian $\boldsymbol{L}$. A slightly different interpretation of equation 2 is that the score for each feature is equal to a weighted sum of its correlation with the eigenvectors, such that

$$s_m = \sum_{i=1}^{n} \lambda_i (\boldsymbol{f}_m^T \boldsymbol{v}_i)^2.$$

A potential drawback of the Laplacian score is its dependence on all eigenvectors, which may reduce its stability in measuring a feature's importance to the data's main structures. To overcome this limitation, Zhao and Liu (2007) derived an alternative score based only on a feature's correlation to the leading Laplacian eigenvectors. A related, more sophisticated approach is Multi-Cluster Feature Selection (MCFS) (Cai et al., 2010), which computes the solutions to the generalized eigenvector problem $\boldsymbol{L}\boldsymbol{v} = \lambda \boldsymbol{D}\boldsymbol{v}$. The leading eigenvectors are then used as pseudo-labels for a regression task with $l_1$ regularization. Specifically, MCFS applies Least Angle Regression (LARS) (Efron et al., 2004) to obtain, for each leading eigenvector $\boldsymbol{v}_i$, a sparse vector of coefficients $\boldsymbol{\beta}^i \in \mathbb{R}^p$. The feature score is set as the maximum over the absolute values of the coefficients computed from the leading eigenvectors, $s_j = \max_i |\beta_j^i|$. The output of MCFS is the set of features with the highest score. In the next section, we derive Spectral Self-supervised Feature Selection (SSFS), which improves upon the MCFS algorithm in several critical aspects.

## 3 Spectral Self-supervised Feature Selection

### 3.1 Rationale

As its title suggests, MCFS aims to uncover features that separate clusters in the data. Let us consider an ideal case where the observations are partitioned into $k$ well-separated clusters, denoted $A_1, \ldots, A_k$, such that the weight matrix $W_{ij} = 0$ if $\boldsymbol{x}_i, \boldsymbol{x}_j$ are in separate clusters. Let $\boldsymbol{e}^i$ denote an indicator vector for cluster

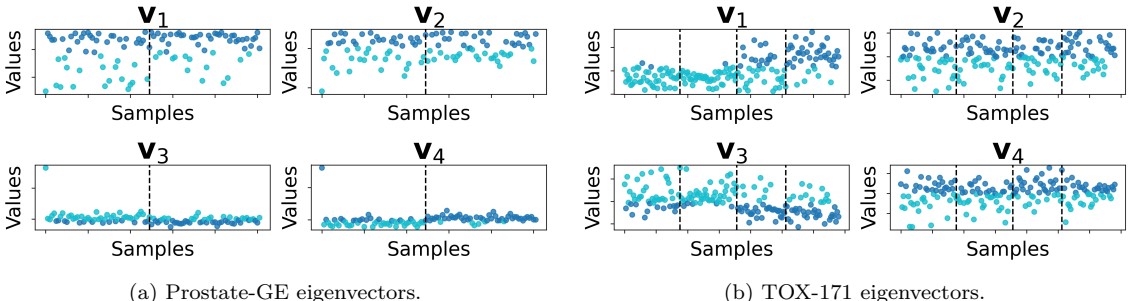

(a) Prostate-GE eigenvectors.                    (b) TOX-171 eigenvectors.

Figure 2: The first four Laplacian eigenvectors of two real datasets. Samples are sorted according to the class label and colored by the outcome of a one-dimensional $k$-medoids per eigenvector. The vertical bar indicates the separation between the classes. In Prostate-GE, $\boldsymbol{v}_4$ is the most informative to the class labels, and an outlier can be seen on the upper left in the third and fourth eigenvectors. In TOX-171, $\boldsymbol{v}_3$ is most informative to the class labels than $\boldsymbol{v}_2$.

$i$ such that

$$
e_j^i = \begin{cases} 1/\sqrt{|A_i|} & j \in A_i \\ 0 & \text{otherwise,} \end{cases}
$$

where $|A_i|$ denotes the size of cluster $A_i$. In this scenario, the zero eigenvalue of the graph Laplacian has multiplicity $k$, and the corresponding eigenvectors are equal, up to a rotation matrix, to a matrix $E \in \mathbb{R}^{n \times d}$ whose columns are equal to $\boldsymbol{e}^1, \ldots, \boldsymbol{e}^k$. In this ideal setting, the $k$ leading eigenvectors are indeed suitable for use as pseudo-labels for the feature selection task. Thus, the MCFS algorithm should provide highly informative features in terms of cluster separation.

In most applications, however, cluster separation is far from perfect, and the use of leading eigenvectors may be suboptimal. Here are some common scenarios: 1) High-dimensional datasets may contain substructures in top eigenvectors, while the main structure of interest appears deeper in the spectrum. For illustration, consider the MNIST dataset visualized via t-SNE in Figure 1a. The data contains images of $3, 6$ and $8$. Figure 1b shows the elements of the six leading eigenvectors of the graph Laplacian matrix, sorted by their corresponding digits. The leading eigenvector shows a clear gap between images of digit 6 and the rest of the data. However, there is no clear separation between digits 3 and 8. Indeed, the next eigenvector is not associated with such a separation. Applying feature selection with this eigenvector may produce spurious features irrelevant to separating the two digits. This scenario is prevalent in the real datasets used in the experimental section. For example, Figure 2a shows four eigenvectors of a graph computed from observations containing the genetic expression data from prostate cancer patients and controls (Singh et al., 2002). The leading two eigenvectors, however, are not associated with the patient-control separation.

2) The leading eigenvectors may be affected by outliers. For example, an eigenvector may indicate a small group of outliers separated from the rest of the data. This phenomenon can also be seen in the third and fourth vectors of the Prostate-GE example in Figure 2a. While the fourth eigenvector separates the categories, it is corrupted by outliers and, hence, unsuitable for use as pseudo-labels in a classical regression task, as it might highlight features associated with the outliers.

3) The relation between important features and the separation of clusters may be highly non-linear. In such cases, applying linear regression models to obtain feature scores may be too restrictive.

Motivated by the above scenarios, we derive Spectral Self-supervised Feature Selection (SSFS). We explain our approach in detail in the following two sections.

### 3.2 Eigenvector processing and selection

**Generating binary labels.** Given the Laplacian eigenvectors $\boldsymbol{V} = (\boldsymbol{v}_1, ..., \boldsymbol{v}_d)$, our goal is to generate pseudo-labels that are highly informative to the cluster separation in the data. To that end, for each eigenvector $\boldsymbol{v}_i$, we compute a binary label vector $\boldsymbol{y}_i^*$ (pseudo-labels) by applying a one-dimensional $k$-medoids algorithm (Kaufman and Rousseeuw, 1990) to the elements of $\boldsymbol{v}_i$. In contrast to $k$-means, in $k$-medoids, the cluster centers are set to one of the input points, which makes the algorithm robust to outliers. In Figure 2, the eigenvectors are colored according to the output of the $k$-medoids. After binarization, the fourth eigenvector of the Prostate-GE dataset is highly indicative of the category. The feature selection is thus based on a classification rather than a regression task, which is more aligned with selecting features for clustering. In Section 5.2 we show the impact of the binarization step on multiple real-world datasets.

**Eigenvector selection.** Selecting $k$ eigenvectors according to their eigenvalues may be unstable in cases where the eigenvalues exhibit a small spectral gap. We derive a robust criterion for selecting informative eigenvectors that is based on the stability of a model learned for each vector. Formally, we consider a surrogate model $h : \mathbb{R}^p \to \mathbb{R}$, and a feature score function $\boldsymbol{s}(h) \in \mathbb{R}^p$, where $p$ denotes the number of features. The feature scores are non-negative and their sum is normalized to one. For example, $h$ can be the logistic regression model $h(\boldsymbol{x}) = \sigma(\boldsymbol{\beta}^T \boldsymbol{x})$. In that case, a natural score function is the absolute value of the coefficient vector $\boldsymbol{\beta}$. For each eigenvector $\boldsymbol{v}_i$, we train a model $h_i$ on $B$ (non-mutually exclusive) subsets of the input data $\boldsymbol{X}$ and the pseudo-labels $\boldsymbol{y}_i^*$. We then estimate the variance of the feature score function, for every feature $m \in \{1, ..., p\}$:

$$\widehat{\mathrm{Var}}(s_m(h_i)) = \frac{1}{B-1} \sum_{b=1}^{B} (s_m(h_{i,b}) - \bar{s}_m(h_i))^2.$$

This procedure is similar (though not identical) to the Delete-d Jackknife method for variance estimation (Shao and Wu, 1989). We keep, as pseudo-labels, the $k$ binarized eigenvectors with the lowest sum of variance, $\hat{\mathcal{S}}_i = \sum_{m=1}^{p} \widehat{\mathrm{Var}}(s_m(h_i))$. We denote the set of selected eigenvectors by $I$. A pseudo-code for the pseudo-labels generation and eigenvector selection appears in Algorithm 1.

### 3.3 Feature selection

For the feature selection step, we train $k$ models, denoted $\{f_i \mid i \in I\}$, to predict the selected binary pseudo-labels based on the original data. Similarly to the eigenvector selection step, each model is associated with a feature score function $\boldsymbol{s}(f_i)$. The features are then scored according to the following maximum criterion,

$$\mathrm{score}(m) = \max_{i \in I} s_m(f_i).$$

Finally, the features are ranked by their scores, and the top-ranked features are selected for the subsequent analysis. The choice of model for this step can differ from that used in the eigenvector selection step, allowing for flexibility in the modeling approach (see Section 3.4 for details). Pseudo-code for SSFS appears in Algorithm 2.

### 3.4 Choice of Surrogate Models

Our algorithm is compatible with any supervised model capable of providing feature importance scores. We combine the structural information from the graph Laplacian with the capabilities of various supervised models for unsupervised feature selection. Empirical evidence supports the use of more complex models such as Gradient-Boosted Decision Trees for various complex, real-world datasets (McElfresh et al., 2023; Chen and Guestrin, 2016). These models are capable of capturing complex nonlinear relationships, which we leverage by training them on pseudo-labels derived from the Laplacian's eigenvectors. For example, for eigenvector selection, one can use a simple logistic regression model for fast training on the resampling procedure and a more complex gradient boosting model such as XGBoost (Chen and Guestrin, 2016) for the feature selection step.

---

**Algorithm 1** Pseudo-code for Eigenvector Selection and Pseudo-labels Generation

---

**Input:** Dataset $\boldsymbol{X} \in \mathbb{R}^{n \times p}$ (with $n$ samples and $p$ features), number of eigenvectors to select $k$, number of eigenvectors to compute $d$, surrogate models $H = \{h_i \mid i \in [d]\}$, feature scoring function $\boldsymbol{s} : \mathcal{F} \to \mathbb{R}^p$, number of resamples $B$

1: Initialize an empty list for the pseudo-labels $\mathcal{Y}^*$ and an empty list for the sums of features variance $\hat{\mathcal{S}}$
2: Compute the significant $d$ eigenvectors of the Laplacian of $\boldsymbol{X}$: $\boldsymbol{V} = (\boldsymbol{v}_1, ..., \boldsymbol{v}_d)$
3: **for** $i = 1$ to $d$ **do**
4:     Binarize the eigenvector $\boldsymbol{v}_i$ using $k$-medoids to obtain $\boldsymbol{y}_i^*$, and append to $\mathcal{Y}^*$
5:     **for** $b = 1$ to $B$ **do**
6:         Subsample $((\boldsymbol{X})_b, (\boldsymbol{y}_i^*)_b)$ from $(\boldsymbol{X}, \boldsymbol{y}_i^*)$
7:         Fit the model $h_{i,b}$ to $((\boldsymbol{X})_b, (\boldsymbol{y}_i^*)_b)$
8:         Compute the feature scores $s_m(h_{i,b})$
9:     **end for**
10:     **for** $m = 1$ to $p$ **do**
11:         Estimate the variance of the $m$-th feature score:

$$\widehat{\mathrm{Var}}(s_m(h_i)) = \frac{1}{B-1} \sum_{b=1}^{B} (s_m(h_{i,b}) - \bar{s}_m(h_i))^2$$

12:     **end for**
13:     $\hat{\mathcal{S}}_i = \sum_{m=1}^{p} \widehat{\mathrm{Var}}(s_m(h_i))$
14:     $\hat{\mathcal{S}} \leftarrow \hat{\mathcal{S}} \cup \{\hat{\mathcal{S}}_i\}$
15: **end for**
16: Select the indices of the $k$ smallest elements in $\hat{\mathcal{S}}$ and store in $I$
17: **return** $\mathcal{Y}^*$, $I$

---

**Algorithm 2** Pseudo-code for Spectral Self-supervised Feature Selection (SSFS)

---

**Input:** Dataset $\boldsymbol{X} \in \mathbb{R}^{n \times p}$ (with $n$ samples and $p$ features) number of eigenvectors to select $k$, number of eigenvectors to compute $d$, surrogate eigenvector selection models $H = \{h_i \mid i \in [d]\}$, surrogate feature selection models $F = \{f_i \mid i \in [d]\}$, feature scoring function $\boldsymbol{s} : \mathcal{F} \to \mathbb{R}^p$, number of resamples $B$, number of features to select $\ell$.

1: Apply Algorithm 1 to obtain the pseudo-labels and the selected eigenvectors:
   $\mathcal{Y}^*, I = \textbf{EigenvectorSelection}(\boldsymbol{X}, k, d, H, \boldsymbol{s}, B)$
2: **for** $i$ in $I$ **do**
3:     Fit the model $f_i$ on $(\boldsymbol{X}, \boldsymbol{y}_i^*)$
4:     Calculate the feature scores $\boldsymbol{s}(f_i)$
5:     Normalize the feature scores such that their sum is one
6: **end for**
7: **for** $m = 1$ to $p$ **do**
8:     Compute the final score for the $m$-th feature:

$$\mathrm{score}(m) = \max_{i \in I} s_m(f_i)$$

9: **end for**
10: **return** a list of $\ell$ features with the highest score.

---

## 4 The importance of a proper selection of eigenvectors: analysis of the product manifold model

As described in Section 3.1, the principle of selecting the leading Laplacian eigenvectors as pseudo-labels is motivated by the case of highly separable clusters, where observations in different clusters have very low

connectivity between them. In many cases, the separation between meaningful states (i.e., biological or medical conditions) may not be that clear. To illustrate this point, consider the MNIST example in Figure 1. While images with the digit 6 constitute a cluster, the separation between digits 8 and 3 are not clearly separated. Figure 3a shows a scatter plot of these digits, where each image is located according to its coordinates in the third and fourth eigenvectors. Even when considering the most relevant eigenvectors, there is no clear separation between the digits. Instead, the transition between 3 and 8 is smooth and depends on the properties of the digits.

To provide insight into the importance of eigenvector selection, we analyze a *product of manifold* model. Our analysis is based on results from two research topics: (i) the convergence, under the manifold assumption, of the Laplacian eigenvectors to the eigenfunctions of the Laplace Beltrami operator associated with the manifold, and (ii) the properties of manifold products. We next provide a brief background on these two topics.

## 4.1 Convergence of the Laplacian eigenvectors

In many applications, high dimensional observations are assumed to reside close to some manifold $\mathcal{M}$ with low intrinsic dimensionality, which we denote by $d$. Many papers in recent decades have analyzed the relation between the Laplacian eigenvectors and the manifold structure Von Luxburg et al. (2008); Singer and Wu (2017); García Trillos et al. (2020); Wormell and Reich (2021); Dunson et al. (2021); Calder and Trillos (2022). More formally, let $\boldsymbol{v}_k$ denote the $k$-th eigenvector of the graph Laplacian, and let $g_k$ denote the $k$-th eigenfunction of the Laplace-Beltrami (LB) operator. We usually assume that $g_k$ is normalized such that

$$\int_{\mathcal{M}} g_k(\boldsymbol{x})^2 \mu(\boldsymbol{x}) dV(\boldsymbol{x}) = 1,$$

where $\mu$ is the distribution function over $\mathcal{M}$. Under several assumptions and proper normalization of $g_k$, we have

$$\boldsymbol{v}_k \xrightarrow[n\to\infty]{} g_k(\boldsymbol{X}).$$

where $g_k(X)$ is a vector of size $n$ containing samples of the function $g_k$ at the $n$ rows of the data matrix $\boldsymbol{X}$.

Let us consider a simple example. Suppose we have $n$ points sampled uniformly at random over the interval $[0, 1]$. The LB operator over this interval is defined as the second derivative, whose eigenfunctions are the harmonic functions given by $g_k(x) = \cos(k\pi x)$. Figure 3 shows the three Laplacian eigenvectors computed with $n = 10^2, 10^3$ and $3 \cdot 10^3$ points. As $n \to \infty$, the vector $v_k$ converges to $g_k(\boldsymbol{X})$.

Here, we use a convergence result from Cheng and Wu (2022), derived under the following assumptions: (i) The $n$ observations are generated according to a uniform distribution over the manifold, such that $\mu(\boldsymbol{x})$ equals to a constant $\mu$. (ii) Let $\lambda_k$ denote the eigenvalue associated with the eigenfunction $g_k$. To ensure the stability of the eigenvectors, we assume a spectral gap between the smallest $K$ eigenvalues bounded away from 0 such that,

$$\min_{i=1}^{K-1}(\lambda_{i+1} - \lambda_i) > \gamma > 0.$$

(iii) The graph weights are computed by a Gaussian kernel $\exp(-\|x_i - x_j\|^2/\epsilon_n)$, with a bandwidth $\epsilon_n \xrightarrow[n\to\infty]{} 0^+$ that satisfies $\epsilon_n^{d/2+2} > C_k \frac{\log n}{n}$ for a constant $C_K$. The following result from Cheng and Wu (2022) guarantees the convergence of the Laplacian eigenvectors to samples of the LB eigenfunction.

**Theorem 1 (Theorem 5.4 of Cheng and Wu (2022))** *For $n \to \infty$ and under assumptions (i)-(iii), with probability larger than $1 - 4K^2 n^{-10} - (2K+6)n^{-9}$, the $k$-th eigenvector $\boldsymbol{v}_k$ of the unnormalized Laplacian satisfies*

$$\left\| \boldsymbol{v}_k - \alpha \boldsymbol{g}_k(\boldsymbol{X}) \right\|_2 = \mathcal{O}(\epsilon_n) + \mathcal{O}\left( \sqrt{\frac{\log n}{n\epsilon_n^{d/2+1}}} \right), \qquad k \le K, \tag{3}$$

*where $\|\boldsymbol{v}_k\| = 1$ and $|\alpha| = o(1)$.*

We next address the eigenvector structure in case of a product-manifold model.

## 4.2 The product of manifold model

In a product of two manifolds, denoted $\mathcal{M} = \mathcal{M}_1 \times \mathcal{M}_2$, every point $\boldsymbol{x} \in \mathcal{M}$ is associated with a pair of points $\boldsymbol{x}_1, \boldsymbol{x}_2$ where $\boldsymbol{x}_1 \in \mathcal{M}_1$ and $\boldsymbol{x}_2 \in \mathcal{M}_2$. We denote by $\pi_1(\boldsymbol{x}), \pi_2(\boldsymbol{x})$ the canonical projections of a point in $\mathcal{M}$ to its corresponding points $\boldsymbol{x}_1, \boldsymbol{x}_2$ in $\mathcal{M}_1, \mathcal{M}_2$, respectively. For example, a $2D$ rectangle is a product of two $1D$ manifolds, where $\pi_1(\boldsymbol{x})$ and $\pi_2(\boldsymbol{x})$ select, respectively, the first and second coordinates.

We denote by $g_i^{(1)}(\boldsymbol{x}), g_i^{(2)}(\boldsymbol{x})$ the $i$-th eigenfunction of the LB operator of $\mathcal{M}_1, \mathcal{M}_2$, respectively, evaluated at a point $\boldsymbol{x}$, and by $\lambda_i^{(1)}, \lambda_i^{(2)}$ the corresponding eigenvalues. In a manifold product $\mathcal{M}_1 \times \mathcal{M}_2$, the eigenfunctions are equal to the pointwise product of the eigenfunctions of the LB operator of $\mathcal{M}_1, \mathcal{M}_2$, and the corresponding eigenvalues are equal to the sum of eigenvalues, such that

$$g_{l,k}(\boldsymbol{x}) = g_l^{(1)}(\pi_1(\boldsymbol{x})) \cdot g_k^{(2)}(\pi_2(\boldsymbol{x})) \qquad \lambda_{l,k} = \lambda_l^{(1)} + \lambda_k^{(2)}. \qquad (4)$$

For simplicity, we denote by $\boldsymbol{v}_{l,k}$ the $(l, k)$-th eigenvector of the Laplacian matrix, as ordered by $\lambda_{l,k}$. An example of a product of 2 manifolds is illustrated in Figure 4b. The figure shows the leading eight eigenvectors of the graph Laplacian. The eigenvectors are indexed by the vector $\boldsymbol{b} = [l, k]$. The full details of this example is provided in the next section.

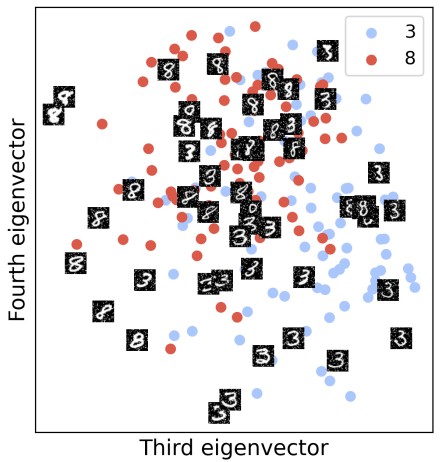
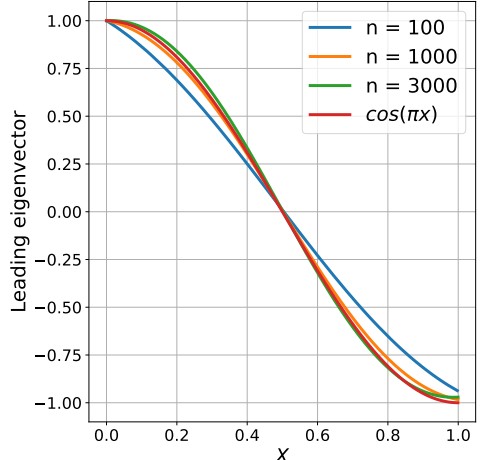

(a) Noisy MNIST: digits 3 and 8

(b) Convergence of eigenvectors for graph on an interval

Figure 3: Panel 3a shows a scatter plot of the noisy MNIST dataset, containing digits 3 and 8, where each image is located according to its coordinates in the third and fourth eigenvectors. Panel 3b shows the leading eigenvector of a graph computed over $n$ points on a $1D$ interval and the leading eigenfunction $\cos(\pi x)$.

## 4.3 Considerations for eigenvector selection in a product-manifold model

We analyze a setting where the $p$ features can be partitioned into $H$ sets according to their dependencies on a set of latent and independent random variables $\theta_1, \ldots, \theta_H$ with some bounded support. A feature $\boldsymbol{f}_m$ that depends on $\theta_h$ consists of samples from a smooth transformation $\theta_h \xrightarrow{F_m} \boldsymbol{f}_m$. We denote by $X^{(h)}$ the submatrix that contains the features associated with $\theta_h$. The smoothness of the transformations implies that the rows of $X^{(h)}$ constitute random samples from a manifold of intrinsic dimension 1.

Figure 4a shows a $3D$ scatter plot, where the axis are three such features with values generated by three polynomials of $\theta_1$. The figure is an illustration of a manifold with a single intrinsic dimension embedded in a $3D$ space. The independence of the latent variables $\theta_h$ implies that the observations $\boldsymbol{x}_i \in \mathbb{R}^p$ are samples from a product of $H$ manifolds, each of dimensionality 1 Zhang et al. (2021); He et al. (2023). The canonical

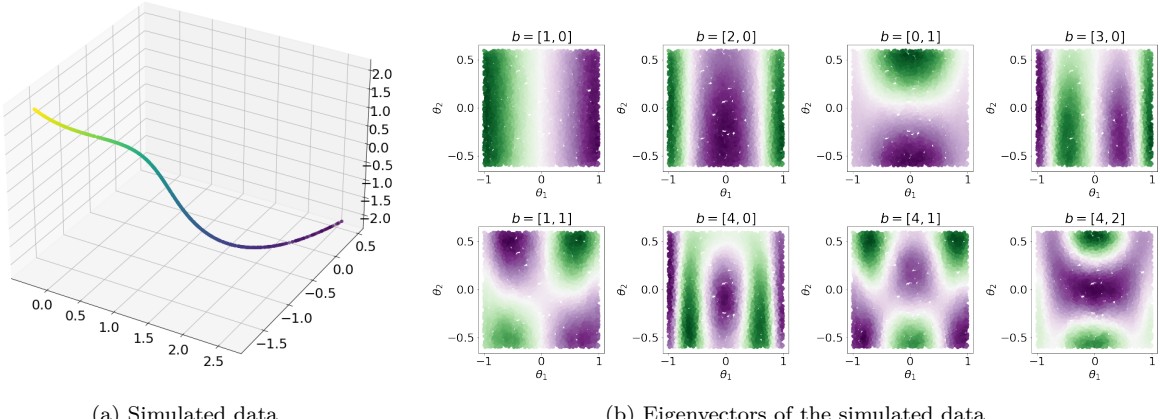

(a) Simulated data          (b) Eigenvectors of the simulated data

Figure 4: Panel (a) illustrates three features of a simulated dataset. Each feature is equal to a different polynomial of the same random latent variable $\theta_1$. Each point in the 3D scatter plot is located according to the values of the three features and colored by the value of $\theta_1$. Panel (b) shows the eigenvectors of the graph Laplacian matrix. Each point is located according to the value of $(\theta_1, \theta_2)$ and colored by the value of its corresponding element in the eight leading eigenvectors. The eigenvectors are indexed by the vector $b$, whose elements $b_i$ determine the eigenvector order in the submanifold $\mathcal{M}^{(i)}$.

projection $\pi^{(h)}(\boldsymbol{x})$ selects the features associated with the latent variable $\theta_h$. According to the eigenfunctions properties in equation 4, the eigenfunctions are equal to the product of $H$ eigenfunctions of the submanifolds $\mathcal{M}^{(h)}$, and can thus be indexed by a vector of size $H$, which we denote by $\boldsymbol{b} \in \mathbb{N}^H$.

$$
g_{\boldsymbol{b}}(\boldsymbol{x}) = \prod_{h=1}^{H} g_{\boldsymbol{b}_h}^{(h)}\big(\pi^{(h)}(\boldsymbol{x})\big) \qquad \lambda_{\boldsymbol{b}} = \sum_{h=1}^{H} \lambda_{\boldsymbol{b}_h}^{(h)}.
$$

Let $\boldsymbol{e}^{(h)}$ denote an index vector with elements $\boldsymbol{e}_j^{(h)} = 1$ if $j = h$ and 0 otherwise. The first eigenfunctions $g_0^{(h)}$ are equal to a constant for all submanifolds $\mathcal{M}^{(h)}$. Thus, The eigenfunctions $g_{\boldsymbol{e}^{(h)}}$ are equal to

$$
g_{\boldsymbol{e}^{(h)}}(\boldsymbol{x}) = g_1^{(h)}\big(\pi^{(h)}(\boldsymbol{x})\big) \prod_{j \neq h}^{H} g_0^{(j)}\big(\pi^{(j)}(\boldsymbol{x})\big) = C g_1^{(h)}\big(\pi^{(h)}(\boldsymbol{x})\big), \tag{5}
$$

where $C$ is some constant. Importantly, the functions $g_1^{(h)}$ and thus $g_{\boldsymbol{e}^{(h)}}$, depend only on the parameter $\theta^{(h)}$. We define by $\mathcal{E}$ the family of vectors in $\mathbb{N}^H$ that include the indicator vectors $e^{(h)}$ or their integer products (e.g. $2e^{(h)}, 3e^{(h)}$ etc.). A similar derivation as in equation 5 shows that for every index vector $\boldsymbol{b} \in \mathcal{E}$, the eigenfunction $g_{\boldsymbol{b}}$ depends on only one of the latent variable in $\theta_1, \ldots, \theta_h$.

**On the relevance of features for choosing eigenvectors as pseudo-labels.** Our goal is to select a set of features that contains at least one (or more) features from each of the $H$ partitions. Such a choice would ensure that the set contains information about all the $H$ latent variables. Clearly, this imposes a requirement on the set of pseudo-label vectors: we would like at least one vector of pseudo-labels that is correlated with each latent variable.

It is instructive to consider the asymptotic case where $n \to \infty$ and hence according to Theorem 1 and the properties of manifold products, the eigenvectors $\boldsymbol{v}_b$ converge to $g_{\boldsymbol{b}}(\boldsymbol{X})$. A proper choice of eigenvectors for pseudo-labels would be the set $\{\boldsymbol{v}_{\boldsymbol{e}^{(h)}}\}_{h=1}^{H}$, as each of these vectors converges to the samples $g_1^{(h)}(\boldsymbol{X})$, and is thus associated with a different latent variable. However, there is no guarantee that these eigenvectors have the smallest eigenvalues.

Consider for example the case for the data illustrated in Figure 4a. Panel (b) shows the leading eight eigenvectors of the graph Laplacian. The leading two eigenvectors are functions of $\theta_1$ and by choosing them we completely disregard $\theta_2$ with an obvious impact on the feature selection accuracy. A better choice for pseudo-labels would be the first and third eigenvectors, indexed by $e_1$ and $e_2$. Therefore, we need an improved criterion for selecting eigenvectors to serve as pseudolabels for the feature selection process. The following theorem, proven in Appendix A.1, implies that the feature vectors $f_i$ are relevant for developing such a criterion.

**Theorem 2** *We assume that the samples are generated according to our specified latent variable model and that assumptions (i)-(iii) are satisfied. Let $f_i \in \mathbb{R}^n$ be a normalized, zero mean feature vector associated with parameter $\theta_h$. Then,*

$$f_i^T v_b = \mathcal{O}(\epsilon_n) + \mathcal{O}\left(\sqrt{\frac{\log n}{n\epsilon_n^{d/2+1}}}\right) \qquad \forall b \notin \mathcal{E}.$$

The theorem is proved via the following two steps. The details of the proof are provided in the appendix.

Step 1: We show that the inner product $f_i^T g_b(X)$ can be written as the inner product of two random vectors with independent elements. Thus, $\left|f_i^T g_b(X)\right|$ is of order $\mathcal{O}\left(1/\sqrt{n}\right)$ by standard concentration inequalities.

Step 2: Combine the convergence of $v_b$ to $g_b(X)$ with the concentration result of step 1.

Theorem 2 implies that one can use the inner products to avoid selecting less informative eigenvectors that depend on more than one variable. Further guarantees, such as selection of a single vector from each variable, require additional assumptions on the feature values, which we do not make here.

In Algorithm 1 we compute the normalized measure of stability for the feature scores $\{s_m(h_i)\}_{m=1}^p$ obtained by the model $h_i$ to predict the labels computed from the $i$-th eigenvector. When the model $h_i$ is linear (or generalized linear), the score is strongly related to the simple inner product of Theorem 2. In that case, Theorem 2 indicates that the inner product between uninformative eigenvectors and all features is close to zero. Thus, we expect the variance (after normalization) to be similar to the variance of random positive noise. The advantage of the stability measure over the simple linear product as an eigenvector selection criterion is that it allows for more flexibility in the choice of model.

## 4.4 Computational Complexity

The computational complexity of our proposed method consists of several key steps. First, constructing the graph representation of the dataset using a Gaussian kernel requires computing pairwise distances between all data points, resulting in a complexity of $O(n^2 p)$, where $n$ is the number of samples and $p$ is the number of features. Following this, the eigendecomposition of the graph has a complexity of $O(n^3)$. This can be reduced by randomized methods to $O(n^2 d)$, since we only require the first $d$ eigenvectors. Once eigenvectors are computed, we apply the $k$-medoids algorithm to cluster the eigenvectors and generate pseudo-labels, adding a further complexity of $O(dkn^2)$, where $k$ is the number of clusters.

After creating pseudo-labels, we use surrogate models such as logistic regression or gradient-boosted decision trees to evaluate the features. Logistic regression has a complexity of $O(nd)$, while gradient-boosted trees, being more complex, scale with $O(Tn \log n)$, where $T$ is the number of trees. We also use a resampling procedure to estimate the variance of feature importance scores. Each resampling involves training a surrogate model, resulting in an overall complexity of $O(BTn \log n)$ for gradient-boosted trees or $O(Bnd)$ for logistic regression. As a result, the overall computational complexity of SSFS is mainly influenced by the graph construction and training the surrogate mode, resulting in a complexity of $O(n^2 d + Bnd)$.

## 5 Experiments

### 5.1 Evaluation on real world datasets

**Data and experiment description.** We applied SSFS to eight real-world datasets from various domains. Table 5 in Appendix F.2 gives the number of features, samples, and classes in each dataset. All datasets are available online [1].

We compare the performance of our approach to the following alternatives: (i) standard Laplacian score (LS) (He et al., 2005), (ii) Multi-Cluster Feature Selection (MCFS) (Cai et al., 2010), (iii) Nonnegative Discriminative Feature Selection (NDFS), (Li et al., 2012), (iv) Unsupervised Discriminative Feature Selection (UDFS) (Yang et al., 2011), (v) Laplacian Score-regularized Concrete Autoencoder (LS-CAE) (Shaham et al., 2022), (vi) Unsupervised Feature Selection Based on Iterative Similarity Graph Factorization and Clustering by Modularity (KNMFS) (Oliveira et al., 2022) and (vii) a naive baseline, where random selection is applied with a different seed for each number of selected features.

For evaluation, we adopt a criterion that is similar to, but not identical to, the one used in prior studies (Li et al., 2012; Wang et al., 2015). We select the top 2, 5, 10, 20, 30, 40, 50, 100, 150, 200, 250, and 300 features as scored by each method. Then, we apply $k$-means 20 times on the selected features and report the average clustering accuracy (along with the standard deviation), computed by (Cai et al., 2011):

$$\text{ACC} = \max_{\pi} \frac{1}{N} \sum_{i=1}^{N} \delta(\pi(c_i), l_i),$$

where $c_i$ and $l_i$ are the assigned cluster and true label of the $i$-th data point, respectively, $\delta(x, y)$ is the delta function which equals one if $x = y$ and zero otherwise, and $\pi$ represents a permutation of the cluster labels, optimized via the Kuhn-Munkres algorithm (Munkres, 1957).

Unlike the evaluation approach taken by Wang et al. (2015); Li et al. (2012), which entailed a grid search over hyper-parameters to report the optimum results for each method, our analysis employed the default hyper-parameters as specified by the respective implementations, including SSFS. This approach aims for a fair comparison to avoid favoring methods that are more sensitive to hyper-parameter adjustments. In addition, it acknowledges the practical constraints in unsupervised settings where hyper-parameter tuning is typically infeasible. Such differences in the approach to hyper-parameter selection could account for discrepancies between the results reported in previous studies and those in our study. See Appendix F.1 for additional details.

Table 1: Average clustering accuracy on benchmark datasets along with the standard deviation. The number of selected features yielding the best clustering performance is shown in parentheses, the best result for each dataset highlighted in bold.

| Dataset | Random | LS | MCFS | NDFS | UDFS | KNMFS | LS-CAE | SSFS |
|---|---|---|---|---|---|---|---|---|
| COIL20 | 65.1±2.1(250) | 61.9±2.4(300) | 67.4±3.3(300) | 63.4±2.6(200) | 61.9±3.5(300) | **68.1±2.0(300)** | 64.2±3.1(30) | 67.1±2.8(300) |
| GISETTE | 70.2±0.1(150) | 70.0±0.0(250) | 70.7±0.0(5) | 58.3±1.9(100) | 69.1±0.1(50) | 54.9±0.0(40) | **70.8±0.0(200)** | 69.7±0.0(150) |
| Yale | 47.8±3.5(250) | 43.9±3.2(300) | 44.4±2.9(300) | 43.5±2.5(250) | 43.8±2.3(50) | 47.2±4.3(300) | 46.2±1.6(10) | **50.3±2.3(100)** |
| TOX-171 | 44.2±1.8(250) | 51.3±1.0(5) | 44.5±0.5(5) | 47.3±0.1(150) | 40.2±3.8(250) | 48.1±3.5(20) | 50.1±5.3(200) | **59.4±2.5(100)** |
| ALLAML | 73.2±1.7(300) | 72.2±0.0(200) | 75.0±0.0(150) | **76.6±0.7(2)** | 66.4±1.3(50) | 59.9±9.2(150) | 63.9±0.0(2) | 75.4±3.2(100) |
| Prostate-GE | 63.0±0.7(30) | 58.8±0.0(2) | 61.8±0.0(100) | 58.8±0.0(2) | 63.6±0.3(50) | 62.7±0.0(50) | 63.7±0.0(40) | **75.9±0.5(10)** |
| ORL | 58.9±1.8(300) | 51.6±1.7(300) | 57.0±2.8(300) | 59.1±2.5(300) | 57.3±2.4(300) | **63.2±2.0(150)** | 61.0±2.0(300) | 61.0±2.2(200) |
| ISOLET | 59.5±1.8(300) | 48.9±2.0(300) | 50.7±1.5(300) | **63.1±2.4(200)** | 44.6±1.7(300) | 52.7±2.3(300) | 63.0±2.6(300) | 59.9±1.4(100) |
| Mean rank | 4.12 | 5.88 | 4.62 | 4.94 | 6.44 | 4.38 | 3.31 | **2.31** |
| Median rank | 4.0 | 6.5 | 5.5 | 5.5 | 6.5 | 4.5 | 2.75 | **2.25** |

Table 1 shows, for each method, the highest average accuracy and the number of features for which it was achieved similarly to (Li et al., 2012; Wang et al., 2015). Figure 5 presents a comparative analysis of clustering accuracy across various datasets and methods, considering the full spectrum of selected features. This comparison aims to account for the inherent variance in each method, addressing a limitation where the criterion of the maximum accuracy over the number of selected features might inadvertently favor methods exhibiting higher variance.

---

[1] https://jundongl.github.io/scikit-feature/datasets.html

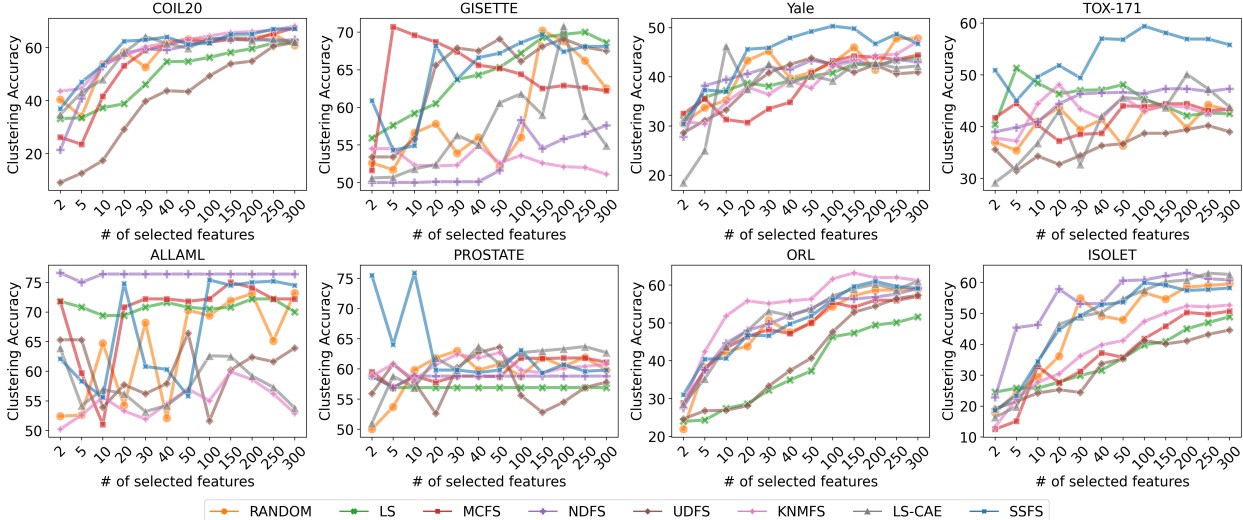

Figure 5: Clustering accuracy vs. the number of selected features on eight real-world datasets.

For SSFS, we use the following surrogate models: (i) The eigenvector selection model $h_i$ is set to Logistic Regression with $\ell_2$ regularization. We use scikit-learn's (Pedregosa et al., 2011) implementation with a default regularization value of $C = 1.0$. Feature scores are equal to the absolute value of the model's coefficients. (ii) The feature selection model $f_i$ is set to XGBoost classifier with *Gain* feature importance. We use the popular implementation by DMLC (Chen and Guestrin, 2016).

Note that we employ the default hyper-parameters for all surrogate models as provided in their widely used implementations. However, it's worth noting that one can undoubtedly leverage domain knowledge to select surrogate models and hyperparameters better suited to the specific domain. For each dataset, SSFS selects $k$ from $d = 2k$ eigenvectors, where $k$ is the number of distinct classes in the data.

**Results.** SSFS has been ranked as the best method in three out of eight datasets. It has shown a significant advantage over competing methods, especially in the Yale, TOX-171, and Prostate-GE datasets. As discussed in Section 3.1, the Prostate-GE dataset has several outliers, and the fourth eigenvector plays a vital role in providing information about the class labels compared to the earlier eigenvectors. SSFS can effectively deal with such challenging scenarios, and this might explain its superior performance. Although our method is not ranked first in the other five datasets, it has produced results comparable to the leading method.

## 5.2 Ablation study

We demonstrate the importance of three SSFS components: (i) eigenvector selection, (ii) self-supervision with nonlinear surrogate models, and (iii) binarization of the Laplacian eigenvectors along with classifiers instead of regressors as surrogate models. The ablation study is performed on a synthetic dataset described in Section 5.2.1, and the eight real datasets used for evaluation in Section 5.1.

### 5.2.1 Synthetic data

We generate a synthetic dataset as follows: the first five features are generated from two isotropic Gaussian blobs; these blobs define the clusters of interest. Additional 45 nuisance features are generated according to a multivariate Gaussian distribution, with zero mean and a block-structured covariance matrix $\boldsymbol{\Sigma}$, such that each block contains 15 features. The covariance elements $\boldsymbol{\Sigma}_{i,j}$ are equal to 0.5 if $i, j$ are in the same block and 0.01 otherwise. We generated a total of 500 samples; see Appendix E.1 for further details. In Figure 6a, you can see a scatter plot of the first five features, and in Figure 6b, you can see a visualization of the covariance matrix. Our goal is to identify the features that can distinguish between the two groups.

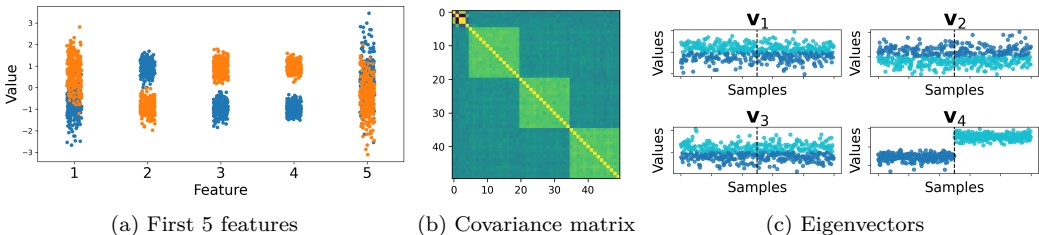

(a) First 5 features      (b) Covariance matrix      (c) Eigenvectors

Figure 6: Visualizations of the synthetic data: Panel (a): scatter plot of the first five features corresponding to the Gaussian blobs, colored by the real label. Panel (b): the covariance matrix of the dataset. Panel (c): the top-4 eigenvectors, samples are sorted by the label and are partitioned by the vertical bar, colored according to the output of $k$-medoids.

Table 2: Ablation study: average clustering accuracy on benchmark datasets, the number of selected features is shown in parenthesis for the best clustering accuracy over the feature range.

| Dataset | no selection | no XGBoost | no selection, regression | regression | SSFS |
|---|---|---|---|---|---|
| COIL20 | 65.0 (150) | 62.1 (150) | **70.5 (100)** | 69.0 (300) | 67.1 (300) |
| GISETTE | **72.5 (10)** | 64.9 (300) | 64.6 (5) | 64.6 (5) | 69.7 (150) |
| Yale | 48.6 (50) | 42.7 (250) | 49.8 (200) | 47.4 (250) | **50.3 (100)** |
| TOX-171 | 50.9 (2) | 45.6 (20) | 45.0 (5) | 45.5 (50) | **59.4 (100)** |
| ALLAML | **75.4 (100)** | 66.7 (50) | 71.1 (300) | 71.1 (300) | **75.4 (100)** |
| Prostate-GE | 59.8 (30) | 69.6 (30) | 61.8 (150) | 61.8 (150) | **75.9 (10)** |
| ORL | 60.0 (300) | 56.8 (300) | 58.5 (300) | 58.5 (200) | **61.1 (200)** |
| ISOLET | 57.0 (150) | 57.1 (300) | **61.3 (300)** | 58.7 (300) | 59.9 (100) |
| Mean rank | 2.94 | 4.0 | 3.0 | 3.5 | **1.56** |
| Median rank | 2.5 | 4.5 | 3.5 | 3.5 | **1.25** |

As Figure 6a demonstrates, the two clusters are linearly separated by three distinct features. Furthermore, examining Figure 6c reveals that while the fourth eigenvector distinctly separates the clusters, the higher-ranked eigenvectors do not exhibit this behavior. This pattern arises due to the correlated noise, significantly influencing the graph structure. The evaluation of this dataset is performed by calculating the true positive rate (TPR) with respect to the top-selected features and the discriminative features sampled from the two Gaussian blobs. The performance on the real-world datasets is measured similarly to Section 5.1.

Table 3: Synthetic data results: Top-3 selected features (sorted in descending order by rank), along with their TPR (relative to the first five features).

| Method | Top-3 Features | TPR |
|---|---|---|
| SSFS | 2, 9, 19 | 0.3 |
| (no XGBoost) | 4, 3, 2 | 1.0 |
| (no selection) | 43, 30, 49 | 0.0 |
| (regression) | 15, 17, 14 | 0.0 |
| MCFS | 47, 7, 43 | 0.0 |

### 5.2.2 Results

**Eigenvector Selection.** We compare to a variation of SSFS termed SSFS (no selection), where we don't filter the eigenvectors. We train the surrogate feature selector model on the leading $k$ eigenvectors, with $k$ set to the number of distinct classes in the data. Figure 7b shows that our eigenvector selection scheme provides an advantage in seven out of eight datasets. Similarly to Sec. 5.1, filtering the eigenvectors is especially advantageous on the Prostate-GE dataset, as our method successfully selects the most discriminative eigenvectors (see Figure 2a ). On the synthetic dataset, the selection procedure provides a large advantage,

as seen in Table 3. Figure 6c illustrates that the fourth eigenvector is the informative one with respect to the Gaussian blobs. Indeed, the fourth eigenvector and the third eigenvector are selected by the selection procedure. This eigenvector yields better features than MCFS and SSFS (no selection), which rely on the top two eigenvectors.

**Classification and regression.** We compare the following regression variants of SSFS , which use the original continuous eigenvectors as pseudo-labels (without binarization): (i) SSFS (regression): uses ridge regression for eigenvector selection and XGBoost regression for the feature selection as surrogate models. (ii) SSFS (no selection, regression): uses the top $k$ eigenvectors without binarization and XGBoost regression. Figure 7a and Table 2 show that SSFS performs best on six of the eight real-world datasets. Interestingly, when using continuous regression as a surrogate model, the selection procedure does not seem to provide an advantage compared to no selection.

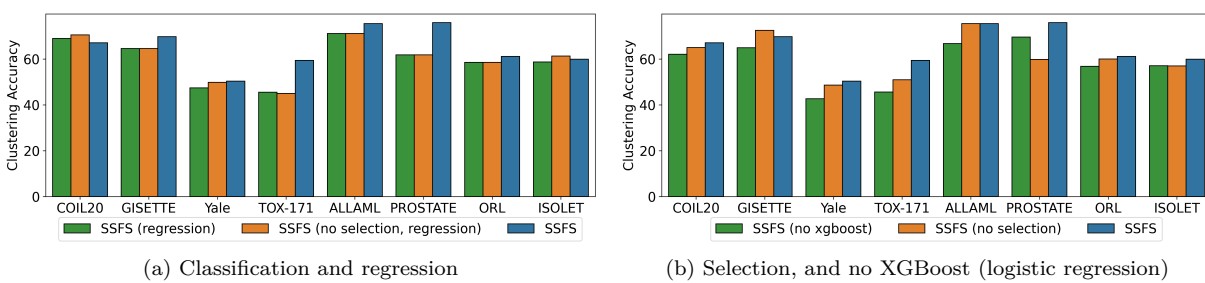

(a) Classification and regression

(b) Selection, and no XGBoost (logistic regression)

Figure 7: Ablation study results on the real-world datasets. The best clustering accuracy over the number of selected features is shown for each method.

**Complex nonlinear models as surrogate models.** We compare SSFS to a variant of our method denoted SSFS (no XGBoost), which employs a logistic regression instead of XGBoost as the surrogate feature selector model. Figure 7b shows that XGBoost provides an advantage compared to the linear model on real-world datasets. On the synthetic dataset, the linear variant provides better coverage for the top-3 features that separate the Gaussian blobs, compared to XGBoost (see Table 3 and Figure 6a). That is not surprising since, in this example, the cluster separation is linear in each informative feature. We note, however, that the top-ranked feature by SSFS with XGBoost is a discriminative feature for the clusters in the data (see Figure 6a); therefore, its selection can still be considered successful in the case of a single feature selection.

## 6   Discussion and future work

We proposed a simple procedure for filtering eigenvectors of the graph Laplacian and demonstrated that its application could have a significant impact on the outcome of the feature selection process. The selection is based on the stability of a classification model in predicting binary pseudo-labels. However, additional criteria, such as the accuracy of a specific model or the overlap of the chosen features for different eigenvectors, may provide information on the suitability of a specific vector for a feature selection task. We also illustrated the utility of expressive models, typically used for supervised learning, in unsupervised feature selection. Another direction for further research is using self-supervised approaches for *group feature selection* (GFS) for single modality (Sristi et al., 2022) or multi-modal data (Yang et al., 2023; Yoffe et al., 2024). In contrast to standard feature selection where the output is sparse, GFS aims to uncover groups of features with joint effects on the data. Learning models based on different eigenvectors may provide information about group effects with potential applications such as detecting brain networks in Neuroscience and gene pathways in genetics.

**Broader Impact Statement**

This work involves methods applicable to high-dimensional data, including sensitive domains like healthcare. While our experiments used publicly available datasets, minimizing immediate privacy concerns, practitioners applying this method to sensitive data should ensure compliance with data protection regulations and adopt robust privacy-preserving measures.

There is also a risk of misapplication in critical areas such as medical diagnosis or criminal justice, where biased or incorrect feature selection could lead to harmful decisions. To mitigate this, we recommend integrating bias detection and mitigation techniques into the training pipeline to enhance fairness and reliability.

Careful consideration of these ethical and societal implications is essential for the responsible use of this method.

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

## A   Product of manifold perspective.

### A.1   Proof of Theorem 1.

As mentioned in the main text, the theorem is proven with the following two main steps:

Step 1: Prove that the inner product $\left| \boldsymbol{f}_i^T \boldsymbol{g_b}(\boldsymbol{X}) \right|$ is of order $\mathcal{O}\left(1/\sqrt{n}\right)$ for all eigenfunctions $\boldsymbol{g_b}(\boldsymbol{X})$ indexed by a vector $\boldsymbol{b} \notin \mathcal{E}$.

Step 2: Combine the result of step 1 with the convergence guarantees in Theorem 1 to bound the inner product $\boldsymbol{f}_i^T \boldsymbol{v_b}$.

**Step 1:**   According to our model, feature $i$ is equal to a smooth transformation of a single latent variables. Assume w.l.o.g that the single variable is $\theta_1$ such that $\boldsymbol{f}_i = F_i(\theta_1)$. By the product of manifold assumption, the eigenfunction $g_{\boldsymbol{b}}$ is equal to

$$\boldsymbol{g_b}(\boldsymbol{x}) = \prod_{h=1}^{H} \boldsymbol{g_{b_h}}(\pi^{(h)}(\boldsymbol{x})) = \boldsymbol{g_{b_1}}(\pi^{(1)}(\boldsymbol{x})) \prod_{h=2}^{H} \boldsymbol{g_{b_h}}(\pi^{(h)}(\boldsymbol{x})).$$

Let $\otimes$ denote the Hadamard product. We can write the inner product $\boldsymbol{f}_i^T \boldsymbol{g_b}(\boldsymbol{X})$ as,

$$\boldsymbol{f}_i^T \boldsymbol{g_b}(\boldsymbol{X}) = \left( \boldsymbol{f}_i \otimes g_{\boldsymbol{b_1}}(\pi^{(1)}(\boldsymbol{X})) \right)^T \left( \boldsymbol{g_{b_2}}(\pi^{(2)}(\boldsymbol{X})) \otimes, \dots, \otimes \boldsymbol{g_{b_H}}(\pi^{(H)}(\boldsymbol{X})). \right. \tag{6}$$

The vectors $\boldsymbol{f}_i$ and $\boldsymbol{g_{b_1}}(\pi^{(1)}(\boldsymbol{X}))$ both depend on $\theta_1$ only. The vectors $\{\boldsymbol{g_{b_h}}(\pi^{(h)}(\boldsymbol{X})\}_{h=2}^{H}$ depend, respectively, on $\theta_2, \dots, \theta_H$. We set

$$\boldsymbol{a}(\theta_1) = \boldsymbol{f}_i \otimes g_{\boldsymbol{b_1}}(\pi^{(1)}(\boldsymbol{X})) \qquad \boldsymbol{d}(\theta_2, \dots, \theta_H) = \boldsymbol{g_{b_2}}(\pi^{(2)}(\boldsymbol{X})) \otimes, \dots, \otimes \boldsymbol{g_{b_H}}(\pi^{(H)}(\boldsymbol{X})).$$

The elements of the random vectors $\boldsymbol{a}(\theta_1)$ and $\boldsymbol{d}(\theta_2, \dots, \theta_H)$ are statistically independent. In addition, we have that $\|\boldsymbol{f}_i\| = 1$ and

$$\|\boldsymbol{g}_h(\pi^{(h)}(\boldsymbol{X}))\| = 1 + o(1) \qquad \forall(h),$$

see for example (Cheng and Wu, 2022, Lemma 3.4). This implies that both $\boldsymbol{a}(\theta_1)$ and $\boldsymbol{d}(\theta_2, \dots, \theta_H)$ are bounded by $1 + o(1)$. The inner product between two independent random vectors with unit norm and iid elements is of order $\mathcal{O}(1/\sqrt{n})$, (see for example (Vershynin, 2020, Remark 3.2.5)). Thus,

$$|\boldsymbol{f}_i^T \boldsymbol{g_b}(\boldsymbol{X})| = |\boldsymbol{a}(\theta_1)^T \boldsymbol{d}(\theta_2, \dots, \theta_H)| = \mathcal{O}(1/\sqrt{n}).$$

**Step 2:**   By the triangle inequality,

$$|\boldsymbol{f}_i^T \boldsymbol{v_b}| = |\boldsymbol{f}_i^T (\boldsymbol{v_b} - \boldsymbol{g_b}(\boldsymbol{X}) + \boldsymbol{g_b}(\boldsymbol{X}))| \leq |\boldsymbol{f}_i^T (\boldsymbol{v_b} - \boldsymbol{g_b}(\boldsymbol{X}))| + |\boldsymbol{f}_i^T \boldsymbol{g_b}(\boldsymbol{X})|. \tag{7}$$

The first term on the right-hand side of equation 7 can be bounded by the Cauchy-Schwartz inequality and Theorem 1 via:

$$|\boldsymbol{f}_i^T (\boldsymbol{v_b} - \boldsymbol{g_b}(\boldsymbol{X}))| \leq \|\boldsymbol{f}_i^T\| \|\boldsymbol{v_b} - \boldsymbol{g_b}(\boldsymbol{X})\| = \mathcal{O}(\epsilon_n) + \mathcal{O}\left( \sqrt{\frac{\log n}{n \epsilon_n^{d/2+1}}} \right). \tag{8}$$

The second term is bounded by step 1. Since the term in equation 8 dominates $\mathcal{O}(1/\sqrt{n})$ for any $\epsilon_n$, this concludes the proof.

## B   Stability analysis

We perform stability analysis using the Variation of Information (VI) criterion introduced by Meilă (2003), a measure of the distance between two clustering assignments. This criterion can be used to measure the

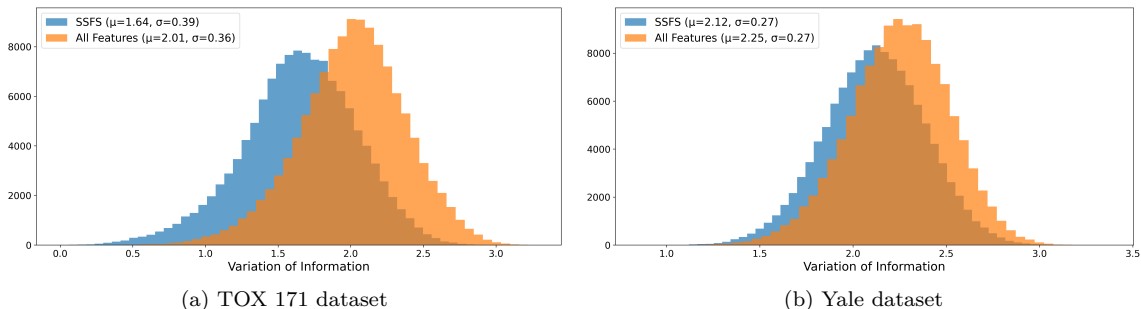

(a) TOX 171 dataset  (b) Yale dataset

Figure 8: Histograms of VI value distributions for SSFS-selected features (orange) and all features (blue) on (a) TOX 171 and (b) Yale datasets. Lower VI values indicate higher stability in clustering results.

stability of clustering results, which is valuable for assessing feature selection quality. For noisy features, we expect unstable clustering outputs with high variability between runs, while informative features should yield more stable results with less variability.

Our analysis involves applying $k$-means clustering to the selected features 500 times with random initializations, calculating the VI between all pairs of clustering outputs, and analyzing the distribution of VI values. Lower average VI and tighter distribution indicate higher stability, suggesting more informative features. We compared the stability of clusters obtained using the original dataset versus those obtained using features selected by our Stable Spectral Feature Selection (SSFS) method.

Table 4 presents results for the TOX171 and Yale datasets, demonstrating clear improvement in stability when using SSFS. For both datasets, SSFS-selected features yield lower mean VI values, indicating more consistent clustering outcomes. Figure 8 visualizes the distributions of VI values for both datasets, showing the shift towards lower VI values when using SSFS-selected features compared to using all features.

Table 4: Stability Analysis Results: Mean Variation of Information (VI) $\pm$ Standard Deviation

| Dataset | SSFS (top 100 features) | All features |
|---|---|---|
| TOX171 | **1.64 $\pm$ 0.39** | 2.01 $\pm$ 0.36 |
| Yale | **2.12 $\pm$ 0.27** | 2.25 $\pm$ 0.27 |

## C  Interpretability

Interpretability is a fundamental aspect of unsupervised feature selection, particularly in domains where understanding the meaning and relevance of selected features is crucial.

In the context of our proposed SSFS algorithm, interpretability can be enhanced in several ways. Feature ranking, by examining the weights or importance scores assigned to features during the selection process, can provide insights into which features are most important across the dataset. Visualization techniques, as demonstrated in Figure 9, can reveal patterns and distributions that highlight the discriminative capability of selected features across different classes or clusters. Domain-specific analysis is another effective approach. In biological datasets, for instance, selected features can be mapped back to specific genes or biomarkers, allowing domain experts to validate the biological relevance of the selected features and potentially uncover new insights about the underlying biological processes.

These interpretability techniques enhance the utility of unsupervised feature selection algorithms like SSFS by providing insights into underlying relationships and feature relevance, which are essential for real-world applications across various domains.

Figure 9 visualizes the top twenty features selected by SSFS on the COIL20 dataset. Each column corresponds to a sample, and each row represents a selected feature. The columns are ordered by the class label. The figure highlights the discriminative effect the selected features have over five classes in this dataset.

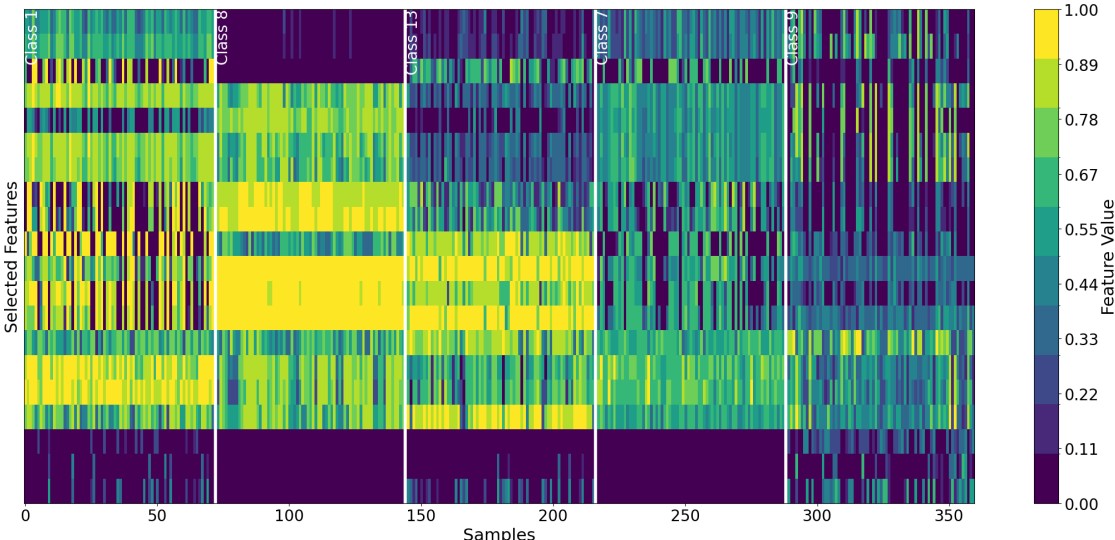

Figure 9: Visualization of the top 20 selected features by SSFS across five classes in the COIL20 dataset. Samples are grouped by class, separated by white vertical lines, with class labels shown at the top. The color intensity indicates the relative feature values. This visualization allows for the comparison of feature patterns and distributions across different classes, highlighting potential discriminative features and class-specific characteristics.

## D    Additional experiments on synthetic data

We conducted further experiments to test our method's robustness to noise, using two synthetic datasets with 100 samples per trial over 10 trials. Both datasets contain five meaningful features from two isotropic Gaussian blobs but differ in their nuisance features. The first dataset's nuisance features are sampled from a multivariate Gaussian distribution, similar to Section 5.2.1, but with varying numbers of nuisance features. The second uses a copula of correlated nuisance features formed of marginals with a Laplace distribution. We measure performance using recall, comparing the top five ranked features to the five known meaningful ones.

Results in Figure 10 show SSFS with logistic regression as a linear classifier (denoted by linear_clf) outperforming other methods across both experiments. For Gaussian nuisance features (Figure 10a), SSFS with a linear classifier maintained consistently high performance up to 60 features, while other methods exhibited a more pronounced decline as the number of nuisance features grew. In the case of Laplace-distributed nuisance features (Figure 10b), the advantage of SSFS with a linear classifier was even more evident, with its performance decreasing more slowly as the number of nuisance features increased. Overall, these experiments demonstrate that SSFS is a robust method for identifying relevant features, performing well for various noise distributions and maintaining high recall even with increasing noise levels.

## E    Ablation study additional details

### E.1    Synthetic data generation

For the synthetic data, we generated 500 samples, where we used the `make_blobs` function from scikit-learn to generate the first five features, with arguments `cluster_std=1`, `centers=2`.

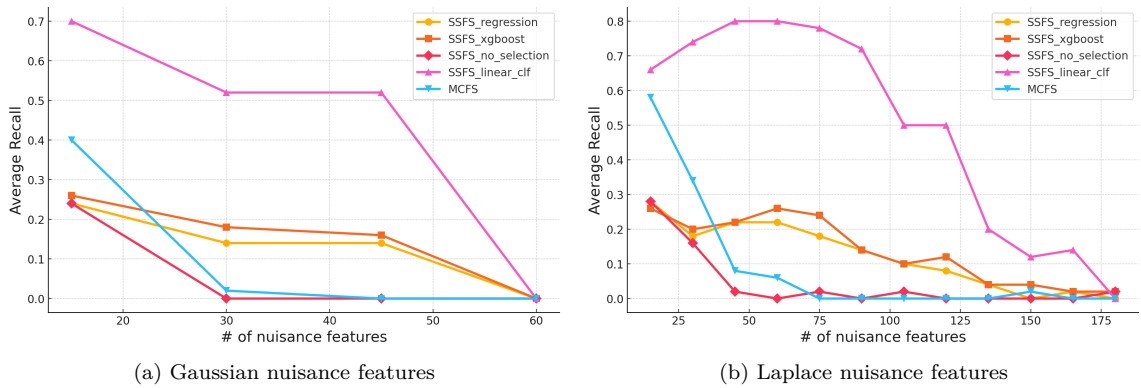

(a) Gaussian nuisance features  (b) Laplace nuisance features

Figure 10: Performance comparison of feature selection methods under increasing noise levels. The x-axis shows the number of nuisance features added to the give meaningful ones, while the y-axis represents the average recall (proportion of the top five ranked features matching the five meaningful features). Each point is the average over 10 trials with 100 samples each. SSFS with a linear classifier (logistic regression) consistently outperforms MCFS and the other SSFS variants.

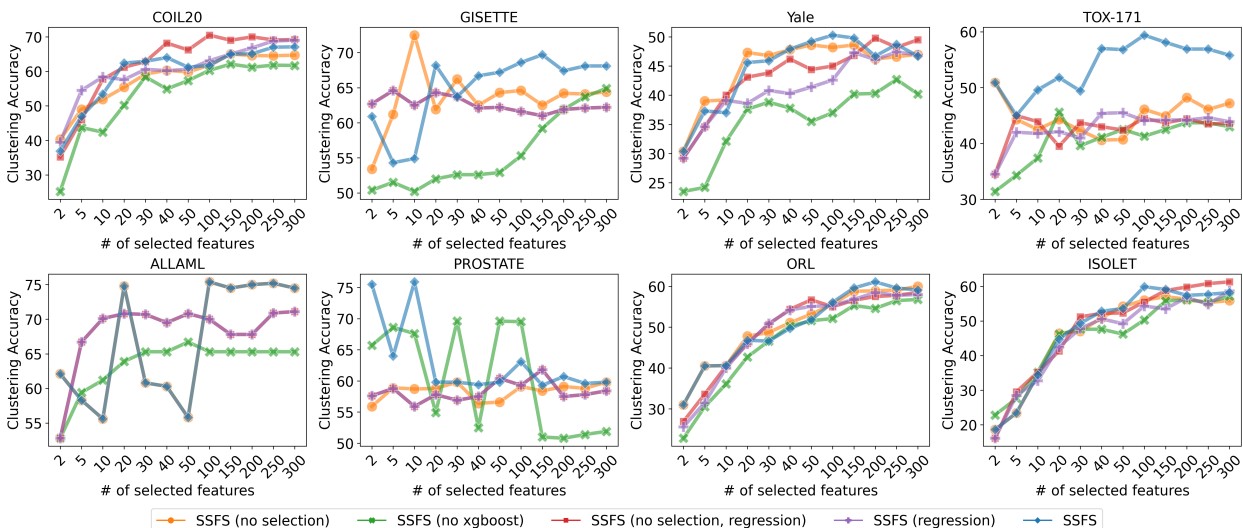

Figure 11: Ablation study: Clustering accuracy on real-world datasets

## E.2 Detailed Experimental Results

In this section, we provide more detailed results of the ablation study. Figure 11 contains comparative analysis in terms of the performance for the whole selected feature range,

# F Experiments on real world datasets additional details

## F.1 Additional Experimental Details

## F.2 Datasets

Table 5 provides information about the real-world datasets used in the experiments.

For all datasets, the features are z-score normalized to have zero mean and unit variance.

Table 5: Real-world datasets description.

| Dataset | Samples | Dim | Classes | Domain |
|---|---|---|---|---|
| COIL20 | 1440 | 1024 | 20 | Image |
| ORL | 400 | 1024 | 40 | Image |
| Yale | 165 | 1024 | 15 | Bio |
| ALLAML | 72 | 7129 | 2 | Bio |
| Prostate-GE | 102 | 5966 | 2 | Bio |
| TOX 171 | 171 | 5748 | 4 | Bio |
| Isolet | 1560 | 617 | 26 | Speech |
| GISETTE | 7000 | 5000 | 2 | Image |

### F.3 Hyperparameters

Hyperparameter selection in unsupervised feature selection presents unique challenges and may be infeasible due to the absence of labeled data for validation. In our methods, we have two types of hyperparameters: (i) For computing the spectral representation (kernel bandwidth and the number of eigenvectors) and (ii) parameters relating to the surrogate model, which predicts the pseudo-labels (i.e., regularization parameter for Logistic Regression). Note that the second type of parameters is used in a prediction task; they can be determined similarly to hyperparameters in supervised settings, for example, through cross-validation.

In our experiments, for the surrogate models, we leverage the default hyperparameters from established machine learning packages such as XGBoost and scikit-learn's logistic regression, as these are generally robust for a wide range of supervised tasks. However, surrogate models and their hyperparameters can also be chosen by performing hyperparameter tuning with cross-validation based on pseudo-labels derived from the eigenvectors. The selection of surrogate models is flexible and can be additionally guided by domain knowledge. Practitioners can also compare results across various models and analyze their selected features.

The first type of hyperparameters is inherent to the method, for example, the number of considered eigenvectors, which should increase with the number of classes and noise level in the data. For instance, in scenarios with more classes or where noise dominates the spectrum, a larger range of eigenvectors should be considered. While we used 500 resamples in our experiments, this number can be adjusted for larger datasets to balance computational cost and performance. Although traditional hyperparameter tuning may not be feasible in most unsupervised scenarios, practitioners working with domain-specific data could potentially fine-tune parameters using cross-validation on similar, labeled datasets from the same domain (such as images).

For SSFS, we use the same hyperparameters, as follows:

- Number of eigenvectors to select $k$ is set to the distinct number of classes in the specific dataset, they are selected from a total of $d = 2k$ eigenvectors.

- Size of each subsample is 95% of the original dataset.

- 500 resamples are performed in every dataset.

- For the affinity matrix, we used a Gaussian kernel with an adaptive scale $\sigma_i \sigma_j$ such that $\sigma_i$ is the distance to the $k = 2$ neighbor of $\boldsymbol{x}_i$.

The Laplacian we used was the symmetric normalized Laplacian.

In the ablation study, for regression, we use scikit-learn ridge regression (for eigenvector selection) and DMLC XGBoost regressor (for the final feature scoring) with their default hyperparameters.

For all of the baseline methods, we used the default hyperparameters. So, for all methods, including SSFS, the hyperparameters are fixed for all datasets (excluding parameters that correspond to the number of features to select and the number of clusters).

For LS, MCFS, UDFS, and NDFS, we used an implementation from the scikit-feature library [2] and inputted the same similarity matrices as SSFS for the methods which accepted such an argument. We fixed a bug in MCFS implementation to choose by the max of the absolute value of the coefficients instead of the max of the coefficients (this improved MCFS performance). For LS-CAE, we used an implementation from [3]. For KNMFS, we used an implementation from [4].

---

[2]https://github.com/jundongl/scikit-feature
[3]https://github.com/jsvir/lscae
[4]https://github.com/marcosd3souza/KNMFS

