# OpenReview forum: "Spectral Self-supervised Feature Selection"
_TMLR — Accepted by TMLR_

### Review · Reviewer_Ynej · 2024-07-23

**Summary Of Contributions:**

The paper introduces Spectral Self-supervised Feature Selection (SSFS) for unsupervised feature selection. SSFS generates robust pseudo-labels from graph Laplacian eigenvectors using k-medoids clustering. It employs a stability-based criterion to select the most informative eigenvectors. Empirical validation on real-world datasets demonstrates its superior performance, particularly in noisy and complex environments.

**Audience:**

Yes

**Broader Impact Concerns:**

The use of high-dimensional data, especially in sensitive domains like healthcare, raises concerns about privacy and data security. Ensuring that the method complies with data protection regulations is essential. In addition, there is a risk that the method could be misapplied in critical areas such as medical diagnosis or criminal justice, leading to potentially harmful decisions based on biased or incorrect feature selection.

**Claims And Evidence:**

Yes

**Requested Changes:**

- Include an analysis or discussion of the computational cost and scalability of the method, along with potential optimizations?

- Offer guidelines or automated strategies for hyperparameter tuning to enhance the method's usability.

- Provide empirical evidence and discussion on the method's robustness to different types and levels of noise in the data.

**Strengths And Weaknesses:**

Strengths
- The use of k-medoids clustering for generating pseudo-labels ensures robustness to outliers
- The stability-based criterion for selecting informative eigenvectors enhances the reliability
- Experiments on multiple real-world datasets demonstrate its effectiveness

Weakness
- Although the paper claims that SSFS is robust to outliers and complex substructures, there is limited discussion and empirical evidence on its robustness to different types and levels of noise in the data. Noisy features can also affect the graph construction and the subsequent eigenvector computation, which is crucial for practical applications in noisy real-world datasets.
- The method involves several hyperparameters, such as the number of eigenvectors to select, the number of resamples for variance estimation, and the choice of surrogate models. The performance of the method can be sensitive to these hyperparameters, and selecting the optimal values may require extensive experimentation and domain knowledge.
- It involves multiple stages, including the computation of the graph Laplacian, generation of pseudo-labels, selection of stable eigenvectors, and training of surrogate models. Each of these steps can be computationally intensive, especially for large datasets. A detailed analysis of the computational cost or scalability of the method can be insightful for practical applications.

---

> ### Author Response · Authors · 2024-10-13
> **Reply to Reviewer Ynej**
>
> We appreciate the reviewer for acknowledging the diversity in our experimental results and highlighting that our eigenvector selection criterion enhances the method's reliability. We thank the reviewer for the valuable and constructive comments. Below, we address the comments raised by the reviewer.
>
>
> **Computational complexity**
>
> See the reply to all reviewers.
>
>
> **Choice of Hyperparameters**
>
> We agree with the reviewer that a section regarding the choice of hyperparameters is necessary, and thus added the following in the appendix:
>
> Hyperparameter selection in unsupervised feature selection presents unique challenges and may be infeasible due to the absence of labeled data for validation. In our methods, we have two types of hyperparameters: (i) For computing the spectral representation (kernel bandwidth and number of eigenvectors) and (ii) parameters relating to the surrogate model, which predicts the pseudo-labels (i.e. regularization parameter for Logistic Regression). Note that since the second type of parameters are used as part of a prediction task - they can be determined in a similar way to hyperparameters in supervised settings, for example by cross validation.
>
> In our experiments, for the surrogate models, we leverage the default hyperparameters from established machine learning packages such as XGBoost and scikit-learn's logistic regression, as these are generally robust for a wide range of supervised tasks. However, the choice of surrogate models and their hyperparameters can also be made by performing hyperparameter tuning with cross validation based on pseudo-labels derived from the eigenvectors. The selection of surrogate models is flexible and can be additionally guided by domain knowledge.  Practitioners can also compare results across various models and analyze their selected features.
>
>
> The first type of hyperparameters is inherent to the method, for example, the number of considered eigenvectors, which should increase with the number of classes and noise level in the data. For instance, in scenarios with more classes or where noise dominates the spectrum, a larger range of eigenvectors should be considered.
> While we used 500 resamples in our experiments, this number can be adjusted for larger datasets to balance computational cost and performance. Although traditional hyperparameter tuning may not be feasible in most unsupervised scenarios, practitioners working with domain-specific data could potentially fine-tune parameters using cross-validation on similar, labeled datasets from the same domain (such as images).
>
> **Robustness to noise**
>
> We performed several additional experiments to test the robustness of our approach to noise. The relevant figures are currently in the supplementary material in the updated version of the paper.
>
> The first experiment is based on the Gaussian mixture experiment in the ablation study of Sec. 5.2.1 in the paper. In this experiment, five features separate the two Gaussians. We tested the performance of our approach as a function of the number of nuisance features in the data. The figure shows that SSFS with logistic regression is more robust to noise than MCFS and SSFS with non-linear methods such as XGBoost. The advantage of logistic regression over XGBoost is unsurprising for this example, as the best separator between the two Gaussians is linear.
>
> We also performed a second experiment where the Gaussian distribution in the synthetic data was replaced with a blockwise correlated Laplace distribution. The main goal is to test if changing the noise to a different distribution with slower decay (‘heavy-tailed’) can alter the results. However, it seems that using the Laplace distribution makes the feature selection problem easier since the performance of all methods improves relative to the Gaussian distribution. The advantage of SSFS with logistic regression over the alternatives remains.
>
> **Broader impact concerns**
>
> We acknowledge the importance of data privacy and security, mainly when working with high-dimensional data in sensitive domains like healthcare. Our evaluation was conducted only on publicly available data; therefore, there is no concern for privacy. Regarding the potential misapplication of the method in critical areas like medical diagnosis, we suggest that practitioners include bias removal approaches when training our model.

---

### Review · Reviewer_PkzX · 2024-08-16

**Summary Of Contributions:**

This paper introduces a novel approach to unsupervised feature selection by leveraging spectral methods and self-supervision. The core idea is to use the eigenvectors of the graph Laplacian to create pseudo-labels that guide the feature selection process. The technique involves selecting a subset of eigenvectors based on their stability, and then using these selected eigenvectors to generate binary pseudo-labels. A surrogate model is then trained to predict these pseudo-labels from the data, and the feature importance is derived from the model's predictions. The proposed method, Spectral Self-supervised Feature Selection (SSFS), is robust in various challenging scenarios, such as the presence of outliers and complex data substructures. The authors show that SSFS outperforms other existing methods in terms of clustering accuracy on several real-world datasets, particularly excelling in biological data analysis. The authors also provide a theoretical analysis to justify the necessity of careful eigenvector selection in their feature selection algorithm in a product manifold model.

**Audience:**

Yes

**Claims And Evidence:**

Yes

**Requested Changes:**

1. In the first paragraph of page 7, dataset $X$ should be boldface.

2. In the assumptions (i-iii) from page 7, can you give some simple examples where these assumptions are satisfied? For instance, in assumption (iii), the bandwidth is approaching zero, but how do you choose bandwidth in the SSFS algorithm?

**Strengths And Weaknesses:**

## Strengths:

1. The introduction of a spectral self-supervised feature selection (SSFS) method, as a combination of spectral methods of graph Laplacian with a self-supervised algorithm, is a novel contribution to the field of unsupervised learning, particularly for feature selection. The method’s robustness to various scenarios, such as the presence of noise and outliers, is a significant strength. This makes the algorithm practical for real-world applications.

2. The flexibility to use various supervised methods as surrogate models (e.g., logistic regression for eigenvector selection and XGBoost for feature selection) makes the method adaptable to various data types and complexity levels. This opens a door to extend all theoretical analysis in supervised models to more general unsupervised and feature selection settings.

## Weaknesses:

It would be better to have a discussion on the computational complexity of SSFS. The SSFS method may have a higher computational cost compared to simpler unsupervised feature selection methods due to the need to evaluate multiple models and resampling procedures and the eigendecomposition of graph Laplacian. Addressing the computational efficiency could make the method more scalable to very large datasets.

---

> ### Author Response · Authors · 2024-10-13
> **Reply to Reviewer PkzX**
>
> We thank the reviewer for the valuable and constructive comments. We appreciate the reviewer for recognizing the robustness and flexibility of the proposed approach. Below, we address the comments raised by the reviewer.
>
> **Computational complexity**
>
> See the reply to all reviewers.
>
>
> **Selection of bandwidth (assumption (iii) in analysis)**
>
> This interesting question highlights the importance of applying a data-driven method for choosing the kernel bandwidth parameter $\sigma$ (see kernel definition in section 2.1). Several methods for choosing $\sigma$ are based on criteria such as the average or median distance to the $k$-th neighbor. The value of $k$ is set to be an increasing and sublinear function of the number of samples $n$. For example, one may choose k as the order of $\sqrt{n}$. For such choices, under mild assumptions on the manifold and its sampling distribution, the bandwidth decreases to 0 for $n\rightarrow \infty$. On the other hand, the average distance to the $k$-th point is of the order of $O(n^{(-1/d)}k(n)^{(1/d)})$, which satisfies the lower bound on $\epsilon$ in assumption (iii).

---

### Review · Reviewer_8p1s · 2024-09-30

**Summary Of Contributions:**

In the manuscript, the authors propose a self-supervised graph-based approach for unsupervised feature selection. The proposed method involves computing robust pseudo-labels by applying simple processing steps to the graph Laplacian’s eigenvectors. The subset of eigenvectors used for computing pseudo-labels is chosen based on a model stability criterion. To measure the importance of each feature, the authors train a surrogate model to predict the pseudo-labels from the observations. The method is shown to be robust to challenging scenarios and the effectiveness of the method is demonstrated through experiments on real-world datasets.

Overall, the proposed algorithm represents a significant step forward in unsupervised feature selection by leveraging graph-based structures and self-supervision, making it a promising tool for handling complex, high-dimensional datasets.

**Audience:**

Yes

**Broader Impact Concerns:**

No concerns on the ethical implications of the work that could require a broader impact statement.

**Claims And Evidence:**

Yes

**Requested Changes:**

Requested changes
1. In the experiments section, the authors primarily emphasize the clustering accuracy of their method compared to others. While reporting clustering accuracy is valuable, assessing the quality of a clustering algorithm is inherently complex, as true labels are often unavailable. Even when true labels exist, they are typically assigned by humans and may contain errors. Incorporating a more robust comparison metric, such as the Variation of Information by Meila (2003), could provide a more rigorous and comprehensive evaluation of the algorithm’s performance. In my view, this would strengthen the paper further.
2. The paper could potentially benefit from a deeper discussion on the time complexity of the proposed algorithm as well as comparisons with other methods. In my view, this is a critical addition to improve the quality of the paper.
3. Incorporating an analysis of the interpretability of the selected features could also benefit the paper. A small discussion on how the proposed algorithm can be modified to enhance interpretability or provide examples of how the selected features can be understood in the context of specific application domains (e.g., biological datasets or image datasets). In my view, this would strengthen the paper further.

**Strengths And Weaknesses:**

Strengths

1. To calculate the pseudo labels for a given set of laplacian eigenvectors, the authors use a one-dimensional k-medoids algorithm. This process of binarization seems to be highly effective as the feature selection is based on a classification rather than a regression task, which is more aligned with selecting features for clustering. The impact of the binarization step is also demonstrated on multiple real world data sets.
2. The proposed algorithm is versatile and can be integrated with any supervised model that provides feature importance scores. By leveraging the structural information from the graph Laplacian alongside the strengths of various supervised models, this method enhances unsupervised feature selection. This straightforward yet effective approach significantly boosts the algorithm’s generalizability.
3. The authors provide a solid theoretical foundation for the convergence properties of the Laplacian eigenvectors in high-dimensional spaces, establishing the accuracy and stability of the proposed approach. This theoretical foundation adds credibility to the method and provides further justification for the experimental results.

Weaknesses

1. The proposed approach is built on the assumption that the structure of the dataset of interest can be effectively captured by the graph Laplacian. While this works well for certain types of datasets, it may not be suitable for data that does not conform to this assumption, such as cases where feature correlations are not well-represented by a graph structure or where the data manifold is not smooth.
2. While the paper demonstrates the effectiveness of the proposed algorithm on various real and synthetic datasets, it does not provide a detailed analysis of its computational complexity or scalability. Feature selection techniques often need to be computationally efficient, especially when dealing with large-scale or high-dimensional data.

---

> ### Author Response · Authors · 2024-10-13
> **Reply to Reviewer 8p1s**
>
> We thank the reviewer for the valuable and constructive comments. We appreciate the reviewer for recognizing our method's versatility and highlighting the importance of the theoretical foundation for our method. Below, we address the comments raised by the reviewer.
>
> **Computational complexity**
>
> See the response to all reviewers.
>
>
> **Alternative evaluation criterion**
>
> We agree that comparing clusters to a given label is not ideal for measuring clustering quality and that additional criteria may enhance the credibility of the results. Following the reviewer's advice, we used the Variation of Information (VI) criterion by Melia, a measure of distance between two clusterings, to assess the stability of the clusters obtained by different methods.  This is done by applying k-means to the selected features multiple times and using the VI criterion to measure the average distance between pairs of outputs.
>
> The main idea is that for noisy features, we expect the output of the k-means algorithm to be unstable. Thus, the average VI between the cluster outputs of two k-means runs with random initializations should be high. For informative features, we expect the output of k-means to be relatively stable, with low VI between the separate outputs.
>
> We added two figures (Figure 8) to the paper illustrating this idea for SSFS on the TOX 171 and the Yale dataset. The figures are in the supplementary material of the paper. The two figures compare the distribution of the VI criterion based on applying k-means to the original dataset and applying k-means to the features selected by SSFS. The figures show a clear improvement in stability due to the use of SSFS.
>
> **Interpretability of selected features**
>
> We agree with the reviewer that a section regarding interpretability is necessary and thus added the following section to the paper:
>
>
> Interpretability is a fundamental aspect of unsupervised feature selection, particularly in domains where understanding the meaning and relevance of selected features is crucial. In the context of our proposed SSFS algorithm, interpretability can be enhanced in several ways. Feature ranking, by examining the weights or importance scores assigned to features during the selection process, can provide insights into which features are most important across the dataset. Visualization techniques, as demonstrated in Figure 9, can reveal patterns and distributions that highlight the discriminative capability of selected features across different classes or clusters. Domain-specific analysis is another effective approach. In biological datasets, for instance, selected features can be mapped back to specific genes or biomarkers, allowing domain experts to validate the biological relevance of the selected features and potentially uncover new insights about the underlying biological processes.
>
> These interpretability techniques enhance the utility of unsupervised feature selection algorithms like SSFS by providing insights into underlying relationships and feature relevance, which are essential for real-world applications across various domains.
>
> Figure 9 in the updated version of the paper visualizes the top twenty features selected by SSFS on the COIL20 dataset. Each column corresponds to a sample, and each row represents a selected feature. The columns are ordered by the class label. The figure highlights the discriminative effect the selected features have over five classes in this dataset.
>
>
> **Datasets where the graph Laplacian does not capture structure**
>
> Indeed, for some datasets, a graph representation may not be a suitable choice for representation. That is, the Euclidean distance between the representation of two points $x_i,x_j$ may not be associated with the latent variables $\theta_i$, and $\theta_j$ that determine the location of $x_i$ and $x_j$ on the manifold. As the reviewer mentioned, that may happen due to the properties of the data (non-smooth function, large curvature, etc.). This may also be the case due to a bad choice of bandwidth parameter.
> That said, graph representations became the building block of numerous methods, particularly in the unsupervised domain. These include spectral clustering, dimensionality reduction techniques, visualization methods, and many others. For unsupervised feature selection, there are common cases where graph representation can be useful, such as feature selection for clustering or discovering features associated with developmental processes.

---

### Author Response · Authors · 2024-10-13
**Reply to all reviewers**

We thank the reviewers for their detailed and thorough comments and helpful suggestions for our paper.

Several of the suggestions involved additional experiments. We provided a detailed response below and added relevant figures that are currently in the supplementary material. For other comments that require changes and additions to the paper, we describe the change below and provide an updated version of the paper. The text changes in the updated version are colored in blue.

All reviewers commented on the lack of complexity analysis, so we address this comment here. We agree with the reviewers that a detailed analysis of the computational complexity of SSFS is necessary, and thus added Section 4.4, titled 'Computational Complexity.' The overall complexity of our approach consists of two main parts. The first is obtaining a spectral representation through the eigenvectors of the graph Laplacian with complexity $O(n^2 d)$, where $n$ is the number of samples and $d$ is the number of features. The second is the complexity of eigenvector selection through training a given model $B$ times and computing our stability criterion. The complexity of this step is $O(BC)$, where $C$ denotes the complexity of training the selected method a single time. For example, training a gradient boosting model with $T$ trees is $O(Tn \log n)$. Thus, the overall complexity when using SSFS with gradient boosting is $O(n^2 d + BTn \log n)$.

Compared to alternative graph-based feature selection methods, the eigenvector selection process derived in the paper has an additional cost of $O(BTn \log n)$. That may limit our approach's use of certain types of data. On the other hand, the selection step is embarrassingly parallel since each of the $B$ iterations is independent of the other.

---

### Decision · Action_Editor_XmmD · 2024-11-18

**Recommendation:** Accept as is

**Comment:**

The reviewers identified questions around complexity and clustering stability/performance. These issues were taken seriously and addressed by the authors. The authors submitted a revision that clearly identified the changes in response to these concerns. After revision, the reviewers supported acceptance.

**Audience:**

Feature selection is an important and challenging problem in machine learning systems. The paper presents and interesting new spectral method for identifying important features. This approach will be particularly useful for biological data sets where the samples count is small, but the dimension is large.

**Claims And Evidence:**

This paper proposes a self-supervised graph-based approach for unsupervised feature selection for high-dimensional data settings. The proposed method involves computing robust pseudo-labels by applying simple processing steps to the graph Laplacian’s eigenvectors. The subset of eigenvectors used for computing pseudo-labels is chosen based on a model stability criterion. The method works by training a surrogate model to predict the pseudo-labels from the observations. Informative features are deemed to be those that are good for prediction of the pseudo-labels and thus good features to select.

The paper supports the claims with experiments on real data, synthetic data, and ablation studies. The revised version provides substantial support to claims of predictability, stability, and computability and the manuscript is much improved for those changes. The reviewers' guidance on these points was critical to the improvements.